# Alzheimer’s Disease—Biochemical and Psychological Background for Diagnosis and Treatment

**DOI:** 10.3390/ijms24021059

**Published:** 2023-01-05

**Authors:** Bocwinska-Kiluk Beata, Jelski Wojciech, Kornhuber Johannes, Lewczuk Piotr, Mroczko Barbara

**Affiliations:** 1Department of Psychiatry, University Hospital of Bialystok, 15-269 Bialystok, Poland; 2Department of Biochemical Diagnostics, Medical University of Bialystok, 15-269 Bialystok, Poland; 3Department of Psychiatry and Psychotherapy, UniversitätsKlinikum Erlangen and Friedrich-Alexander Universität Erlangen-Nürnberg, 91-054 Erlangen, Germany; 4Department of Neurodegeneration Diagnostics, Medical University of Bialystok, 15-269 Bialystok, Poland

**Keywords:** Alzheimer’s disease, psychological intervention, diagnosis, treatment

## Abstract

There is a paucity of empirical research on the use of non-pharmacological interventions to both treat and curb the spread of Alzheimer’s disease (AD) across the globe. This paper examines the biochemical and clinical outlook and the social implications of the condition in relation to psychological aspects that may indicate a direction for further interventions. There is a scarcity of research on the effectiveness of using various psychological aspects of AD, a disease characterized by a process of transition from health and independence to a dependent state with a progressive loss of memory and functional skills. The paper investigates the biochemical and psychological aspects of AD and their significance for improving quality of life for patients with this disease. Psychological interventions based on, among other factors, biochemical studies, are conducted to improve the emotional wellbeing of AD patients and may assist in slowing down the progression of the disease. To date, however, no effective methods of AD treatment have been established.

## 1. Clinical and Social View of Alzheimer’s Disease

Statistical data reveal that in the early 21st century, over 35 million people suffered from Alzheimer’s disease (AD), a progressive neurodegenerative disorder which is one of the most common causes of dementia. The prognostic estimates of Alzheimer’s Disease International (ADI) indicate that the number of patients will double over the next 20 years, possibly leading to a significant social crisis requiring the development of a comprehensive dementia plan in order to manage such a considerable increase [1]. In Poland, there are over 300,000 Alzheimer’s sufferers, predominantly among older people [2]. Regrettably, the rising prevalence of AD is not matched by advances in diagnostic methods or an increase in effective treatment options, with available treatment primarily consisting of symptom alleviation. The effects of the disease on both the patient and his or her immediate environment are profound. According to research data published by the Alliance of Polish Alzheimer’s Organizations in 2015, almost 90% of those caring for people with AD are family members—32% are spouses, and 62% are children of the affected individual. In addition to general cognitive decline, problems with abstract reasoning and time–space orientation, the clinical characteristicsof AD may comprise other behavioral and mood elements which are observable in three distinct stages of the disease [3]. AD is associated with the loss of key functions such as memory, language, reasoning, spatial awareness and even mobility. The disease can also manifest itself through psychological symptoms such as delusions, anxiety, depression and even personality changes (Figure 1). Behavioral and mood symptoms are observed in a majority of AD patients and their prevalence is reported to be as follows: 75% of AD patients display disturbing and/or inappropriate behavior related to agitation; 60% show wandering behavior; 50% show depressive symptoms (anhedonia, dysphoria, guilt, decreased appetite, weight loss and loss of concentration), tendency to scream, psychotic symptoms (hallucinations, delirium and paranoia). Screaming and wailing is reported in 25% of AD patients, while aggressiveness and violence is observed in 20% of patients with AD. The behavioral and mood disturbances afflicting patients with AD are a major source of primary caregiver burden and an important contributor to the decision to admit AD patients to institutionalized care [4,5]. The prevalence and severity of these disturbances are patient-specific [6]. Reported differences arise from the degree and location of brain damage, premorbid personality, the environment and relationships with other people (Figure 2).

## 2. Biochemical Diagnosis of Alzheimer’s Disease

Alzheimer’s disease causes the death of basal forebrain cholinergic neurons due to deposits of abnormal proteins which constitute specific histological markers. With the destruction of approximately 80% of these neurons, the brain’s limits of cognitive performance are exceeded. AD is characterized by a loss of neurons, synaptic degeneration and formation of senile plaques from aggregates of amyloid peptides [7]. The histological criterion for the diagnosis of AD is the finding of filamentous degeneration from twisted filaments, whose main componentsare hyperphosphorylated tau protein and neuropilic threads. The disease is marked by the degeneration of four amino messenger systems—serotonergic, noradrenergic, histaminergic and cholinergic—which constitute the N-methyl-D-aspartate (NMDA) receptor system. These receptors appear to signal transmission at the synapses of the hippocampus and are responsible for synaptic plasticity of the brain (Table 1). Furthermore, nicotinic acetylcholine receptors account for many neurophysiological functions of the brain and any loss of them leads to the development of the disease [8]. An important element in the pathogenesis of AD is oxidative damage. The function and structure of the brain is profoundly affected by cytokines and steroid hormones, including estrogen, testosterone and glucocorticoids [9]. Some factors, such as excitotoxicity, oxidative stress, cholinergic dysfunctions, tau protein hyperphosphorylation, changes in amyloid-beta peptide metabolism, apolipoprotein E, herpes viruses, insulin resistance, glycogen synthase kinase 3 and the endocannabinoid system seem to be related to pathophysiology of Alzheimer’s disease. A factor playing a role in the pathogenesis of Alzheimer’s disease may also be electromagnetic radiation affecting neurotransmitters that play an important role in transmitting signals [10,11].

Laboratory diagnosis of AD should be based on direct detection of pathological changes or detection of markers in biological material showing altered expression in the early stage of the disease or which have applications in the differential diagnosis of other neurodegenerative diseases [12]. To date, efforts to establish a reliable test allowing for an early diagnosis of AD have been unsuccessful. An ideal marker should be characterized by ease of identification and non-invasiveness of testing. Such a marker should recognize asymptomatic disease, track disease progression, assess treatment effectiveness and allow for the differentiation of AD from other neurodegenerative diseases. Regrettably, the preclinical stage of AD—the period prior to manifestation of clinical symptoms—may last for a number of years. Detection of brain atrophy in imaging studies (magnetic resonance imaging or computed tomography) is not sufficient to diagnose AD, only being able to exclude a number of other diseases of the central nervous system. The primary goal of AD biomarker research is to equip doctors and laboratory diagnosticians with biochemical or neuroimaging tools that would allow for accurate diagnosis in the early stage of neuronal wasting [13]. Currently employed techniques for identifying individual biomarkers are based on proteomics and metabolomics. Proteome profiling of the cerebrospinal fluid has revealed more than 1500 different proteins, approximately 130 of which were significantly elevated in patients with AD when compared to healthy individuals. The diagnostic indices of a biomarker (sensitivity, specificity, positive and negative predictive values) require evaluation. If a marker is to be validated for diagnosis of AD, it should have a sensitivity and specificity of at least 85% and a positive predictive value of 80%. In the analysis, examination of a single component is invariably applied, while appropriate techniques allow for the simultaneous determination of a variety of substances or compounds, possibly increasing the values of diagnostic indicators of the tests. Measurement of hyperphosphorylated tau (P-tau) allows for differentiation of AD from other dementias with 85% accuracy, while the combined measurement of tau and Aβ1-42 can predict disease progression with 95% sensitivity and 87% specificity [14]. In the field of research on AD, a conflict has existed for many years. The amyloid hypothesis supporters argue thataggregation of amyloid-beta is a major factor in the onset of AD. On the other hand, opponents argue that amyloid deposition is an epiphenomenon that has diverted attention away from the real causes of AD, which are still mostly unknown.Now that amyloid-beta plaque-removing drugs for AD are available, the dispute over the amyloid hypothesis has gone from the lab to the clinic. Many amyloid-beta-targeting drugs have proved to be clinically ineffective. However, FDA has granted accelerated clearance to aducanumab, one of four anti-amyloid-beta antibodies that have been demonstrated to mediate the removal of amyloid plaque from the brains of AD patients. Current therapeutics for tau treatment include targeting multiple forms of intracellular and extracellular tau to prevent pathological tau formation, accumulation and spread. Many other drugs are in development and trials; however, experimental drug trials warrant further discussion.

In recent years, the National Institute of Aging (NIA) and the International Working Group (IWG-2) have developed criteria for the early diagnosis of AD based on biomarkers and imaging data. These criteria are based on the use of a multidisciplinary approach comprising imaging modalities, clinical biochemical tests and the neuropsychological evaluation of the patient. Biochemical and imaging diagnostic criteria relate to the main features of AD—β-amyloid plaques and tau protein tangles (NFTs). A number of molecules capable of penetrating through the blood–brain barrier have been identified; these bind to Aβ or tau NFT plaques, allowing for the assessment of tau aggregates in the human brain (Figure 3). These precursor molecules include Aβ-binding compound B from Pittsburgh, which ushered in the era of AD neuroimaging [15,16]. In order to facilitate interpretation of results and potential inter-facility comparison, an algorithm that enables evaluation of AD biomarkers in the cerebrospinal fluid has been created. On this scale, a score of zero (0) indicates a high probability of the absence of neurochemical disorders, while a score of four(4) signals a high probability of the presence of AD [17,18].

It has been demonstrated in clinical trials and animal studies that inflammatory processes play an important role in AD—something which may well increase a neurological deficit. A number of inflammatory substances that increase the permeability of the blood–brain barrier can be detected in a number of diseases and can be tested for diagnostic utility. In recent years, a plethora of studies investigating changes in the concentrations of peripheral inflammatory molecules in AD have been published. Studies of the Il-1 family of cytokines and their receptors in serum (12 proteins from this family) have revealed enhanced concentrations of these substances inpatients with AD compared to the control group. The authors suggest that there may be a relationship between a response to the inflammatory process and development of the disease [19]. Regrettably, based on current knowledge, we cannot state unequivocally that changes in the concentrations of various chemokines in the peripheral blood are related to the neurodegeneration of the central nervous system. It must also be considered that changes in the levels of these proteins might be due to systemic diseases or, simply, to aging.

The goal of AD research is not only to understand the causes of neurodegenerative disorders, but also to improve existing diagnostic tests and establish new methods that would enable earlier diagnosis of the disease. Despite the long preclinical stage of the disease, currently available treatments only alleviate the symptoms of AD. The diagnosis of the disease is established by employing a multidisciplinary approach. Due to the fact that AD is a heterogeneous and multifactorial disorder, diagnosis ought to be based on analysis of various proteins which reflect numerous pathological mechanisms. Furthermore, it is advisable that the diagnostic process includes both imaging tests and a neuropsychological examination. It is necessary to create a panel of highly specific proteins, the concentrations of which will change according to disease severity. Ideally, these biomarkers should be measured in blood samples, as this is less invasive than collecting cerebrospinal fluid during a lumbar puncture. However, data on new biomarkers are inconsistent due to differences in detection methods, test standardization and the setting of limit values. Therefore, it is vital that computational models that would allow for prediction of the progression of pathology through the use of machine learning (ML) are created. Biomarker studies supply data necessary for the development of a disease model using ML and artificial intelligence (AI), which can improve diagnostic accuracy and provide information on disease progression. Research teams from different fields should join forces to create reliable models of disease. In the future, AI and ML models based on data obtained from biochemical studies of neuroimaging and psychological tests will play an increasingly important role in predicting the course of Alzheimer’s disease [20].

## 3. Psychological Approach to Challenges Associated with Alzheimer’s Disease

From a psychological perspective, several important challenges can be identified throughout the course of AD. These include the unavailability of screening testing and the lack of standardization of screening tests that are recommended in routine medical diagnostics. Significantly, early detection of AD would generate further challenges and psychological problems, a selection of which is briefly outlined below.

One of the major challenges associated with AD is diagnostic disclosure to patients and their families. Who is informed, when they are informed and by whom they are informed still remain important issues to consider, as individuals learning that they are suffering from an incurable disease, with a gradually or rapidly worsening spectrum of disorders, can experience a secondary exacerbation of symptoms following diagnostic differentiation. Notably, symptoms can occur with variable frequency following the diagnosis, with some being prevalent even before the diagnosis is made. It has been indicated that 72% of patients experience depression, mood changes, social withdrawal and suicidal thoughts two years before the AD diagnosis; 45% of patients develop hallucinations, paranoia and delusions in less than a month following diagnosis; and 21% exhibit irritability, agitation and aggressive behavior within a year following diagnosis [21]. It needs to be emphasized that the above-mentioned alterations or disorders can occur at any time during the course of the disease [21,22]. Moreover, they can be related to the period of acceptance following the loss of behavioral and cognitive functioning or alterations to the living environment, the discontinuity in the sense of oneself and one’s loved ones and deteriorating relationships in the family and in the wider social environment [23]. “It is crucial to recognize that the life of the person with dementia and their caregiver will often be very closely entwined, particularly where informal care is being provided by a partner, or by a relative living in the same house. In such circumstances, the interests of the person being cared for and the person providing the care will often be inseparable, and this may have significant implications for how the various needs and interests at stake are balanced and compromises sought” [24].

Another psychologically interesting aspect of AD is the fact that symptoms and the rate at which they develop can be related to an individual’s emotional response to the disease and the response of the patient’s immediate environment, including significant people providing home care or institutional care and who are vulnerable to emotional burnout as a consequence [25,26]. A number of researchers indicate that AD symptoms, which can be described as heterogeneous and fluctuating, are influenced by the environment or aggravated by emotions expressed by the caregivers [4]. Therefore, psychological support for caregivers becomes a challenge for the health and prevention policy implemented in Poland and beyond.

Consequently, an open and ethically debatable question remains of whether increasing the effectiveness of AD biomarker detection years or months prior to the onset of the disease and informing patients about their biomarker-based diagnosis would not accelerate the incidence of the disease due to the emotional state related to:(a)initiation of the period of acceptance following the imminent and inevitable loss of behavioral and cognitive functioning;(b)awareness of the lack of effective treatment;(c)constant increase in cognitive deficits, which is difficult for patients and their families to accept [27].

This is all the more so because the influence of personality and environmental changes on the neurological functioning of patients with AD is apparent: experiencing hostile environmental changes can lead to aggressive behavior, while travelling or trips can exacerbate some cognitive symptoms [28].

On the other hand, early AD detection and diagnostic disclosure can increase patients’ likelihood of arranging issues surrounding care, legal, medical, therapeutic and financial matters for the future, not to mention discussing their wishes regarding these issues with their families or appointing a legal representative to act on their behalf. It is evident that the need to care for a person with AD is not merely confined to addressing their neurobiological deficits. The disease implies a host of difficulties and discomfort that families must confront when dealing with maladaptive behaviors and, therefore, necessitates that interventions aimed at such problems are developed. The interventions include counselling networks and a professional team of psychological, therapeutic, pedagogical, medical and social workers equipped with so-called soft skills that allow for attuning to patients and addressing their needs more fully; from disclosure practices to demonstrating care. Additionally, problematic behaviors exhibited by AD patients are not a direct result of their impairment or degeneration of neurological function. Firstly, they are an individual adaptive strategy to adjust to or face up to the consequences of dementia [6,29,30]. Secondly, they are a response from the environment and its method of coping with changes within the patient [31,32]. Thus, the way patients and their caregivers cope with the disease and the limitations it brings with it amounts to maintaining a relative emotional balance and a positive or negative self-image, the ability to accept (or not) an uncertain future and losing touch with the past, along with adapting to a new care environment, whether it be family or institutional. Additionally, negative emotions are generated by the upheavals for both the patient and their family, by major or less significant changes in the professional environment following the diagnosis, or by the need to develop new relationships with medical staff, care workers and society in general. Sometimes, they are triggered by the iatrogenic nature of prescribed medications used to manage exacerbations.

Another psychological aspect is the instigation of various therapies offering improvements to the patient’s quality of life. Researchers have noted that the pathological process of AD offers a significant amount of freedom and variety in terms of intervention options, with emphasis on the contribution of non-pharmacological methods of treatment [33]. This results in the need to deal with behavioral problems and for psychoanalytic understanding of personality changes occurring in patients [34,35]. In fact, social interest in coping with behavioral problems and emotional aspects of functioning of individuals suffering from AD are given equal weight. Access to and understanding of the patient’s own emotional states and various ways of regulating them, including physiotherapy (improving motor function), environmental therapy (family relations) and individual or group psychotherapy are closely related to the functioning of the patient at all post-diagnosis stages.

Cognitive behavioral therapy has been proven effective for cognitive decline, allowing patients to focus on the present and to develop disease management skills, such as placing notes around the home (e.g.,“clothes are in the closet”) or using a diary or journal to record upcoming tasks. With the gradual decline of short-term memory while long-term memory persists, techniques of depth psychology can be helpful, e.g., transporting the patient back in time through the use of music, sounds, photographs or smells, which puts patients back in touch with their“old life” and provides them with a feeling of greater integration of the image of self and the object alongside a sense of still being socially bonded.

Reminiscence sessions can be conducted individually or in groups, even in a nursing home setting. A new and specific branch is life history therapy, whereby the patient constructs an autobiography, usually with the help of photographs provided by the family. A characteristic feature of all psychological therapies for patients exhibiting symptoms of dementia in AD is the very low likelihood of successful completion of such therapies and so-called “eternal therapy”, as patients’ condition, despite therapeutic interventions, is bound to gradually deteriorate.

“Talking therapy” is an ad hoc prophylaxis aimed at reducing damage and comprises elements of coping with emotions. As previously indicated, there is no effective drug treatment that can prevent the progression of the disease. However, talking therapies can enhance AD patients’ quality of life and are thus encouraged by the Alzheimer’s Society: “People find it understandably hard to make sense of what is happening to them and how their life is changing.Some feel angry, confused, frightened or anxious.Talking therapies may offer someone with dementia the opportunity to speak openly about their feelings and help them to live with their condition more successfully”. Tension may be exacerbated by a decrease in the patient’s sense of independence and an increase in social control. It may also increase with the use of devices designed to improve patient safety (e.g., GPS locator devices, video cameras or automatic pill dispensers), the involvement of institutions aimed at protecting patients’ interests and wellbeing or the need to rely on legal representatives in the decision-making process [36]. Limiting patients from participating in decisions regarding their medical condition and other associated choices (self-determination in terms of dignity and end-of-life care, resuscitation, tube feeding, place of residence, etc.) often requires psychological and medical consultations.

Another challenge in the psychological domain is respect for patients’ dignity and their right to make decisions on various levels of individual and social deficits while making preventive efforts with regard to risks of violations of those rights by other individuals. The moral dilemma related to care affects both AD patients and their caregivers and concerns the administration of drugs without the patient’s consent or knowledge, sometimes resorting to deception (e.g., hiding medication in beverages) and progressive control of the patient.

To sum up, it should be emphasized that AD, due to the increasing prevalence and type of disorders that appear in the clinical picture and social image of the patient, constitutes a special problem, not only medical or social, but also psychological. Support in the form of a broad range of counseling and psychotherapy is what patients simply deserve and is guaranteed under the Constitution of the Republic of Poland (Article 68, Section 1 grants every person the right to health care; Article 38 grants the legal protection of the life of every human being; Article 30 grants the right to protection of human dignity; Article 40prohibits torture or cruel, inhuman or degrading treatment or punishment) and the requirements of a civilized society. As indicated by Sigmund Freud, “The first requisite of civilization … is that of justice”, and “the first human who hurled an insult instead of a stone was the founder of civilization” [37], which he subsequently materialized through action. “Without love, we get sick. … It is that we are never so defenseless against suffering as when we love, never so helplessly unhappy as when we have lost our loved object or its love” [37]. Patients with AD lose access to a coherent representation of themselves and of an object, and their life circumstances change dynamically. This means that they begin to live with a sense of alienation in an alien place, their words lose all meaning and are incomprehensible in modern civilization, which becomes increasingly alien to them. Many activities have a beneficial effect on the mental state of the patient, reducing his or her depressiveness and sense of isolation, and are a source of satisfying essential needs—pleasure, joy, importance and social contacts.These include music therapy, art therapy and exercise therapy.They are a source of stimulation and calming, increase the quality of life and are easy to use at home.

## 4. Discussion

The major problem of the psychological nature associated with AD can be understood as a loss of aspects of self and object, which is accompanied by gradual behavioral and cognitive impairment, emotional difficulties and, frequently, painful separation from loved ones and familiar surroundings.

Due to the fact that a significant increase in the incidence of AD worldwide is predicted over the next several decades, combining pharmacotherapy with a psychological approach is taken into consideration in order to extend the wellbeing and functionality of people diagnosed with AD. Taking into account the ambiguous aspects and the psychological nature of the sociomedical profile of the disease is necessary in the search for proven and evidence-based psychological interventions. These would constitute an affordable treatment method to be used in combination with pharmacotherapy to delay and alleviate the symptoms of the disease. A multicenter, randomized controlled clinical trial conducted by researchers in Australia [38], which aimed to determine whether physical activity reduces the rate of cognitive decline among people with Alzheimer’s disease, proved successful. This somewhat undermines the belief that neurological damage to the brain is the direct and only cause of the patient’s behavior. Focusing on the psychological aspects of the disease and its various therapies as well as strictly psychological interventions which can improve the nervous and emotional state of patients and their caregivers can constitute a critical response in addition to non-pharmacological methods of causal-symptomatic treatment [39].

Available non-pharmacological and pharmacological therapies have shown only modest effects in slowing the progression of dementia. Non-pharmacological intervention in the treatment of dementia patients, the so-called MAKS therapy, was developed in Erlangen. It is a multicomponent group therapy consisting of tasks divided intocategories—motor stimulation (M), ADL (A) and cognition (K)—preceded by a short introduction consisting of a spiritual element. This highly standardized, multi-component, non-drug, group therapy in a nursing home setting was able to delay cognitive decline in demented patients and their ability to carry out daily activities for at least 12 months [40]. The search for effective therapies for at least mild to moderate AD is ongoing [41,42,43].

This review addresses many aspects of AD, including the psychological aspect often overlooked by physicians and the role of a person’s family and caregivers in their overall wellbeing.This is especially important in the absence of an effective treatment for AD. There is a pressing need for a scientific focus on the psychological aspects of effective management of individual patients and the management of AD in ageing populations worldwide. The basis for the diagnosis of AD, both in scientific and clinical trials, should be diagnostic criteria [44]. No such uniform criteria have been developed thus far, but research is still ongoing. Brain changes associated with Alzheimer’s disease begin to develop even 20 years prior to the onset of clinical symptoms of the disease, and it is hoped that pre-symptomatic tests for AD will soon be developed, alongside effective treatments which can be applied at the earliest stage of the disease.

## Figures and Tables

**Figure 1 ijms-24-01059-f001:**
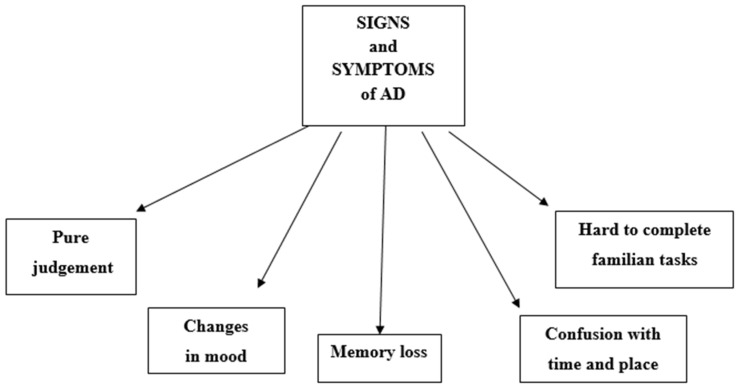
Early symptoms of AD.

**Figure 2 ijms-24-01059-f002:**
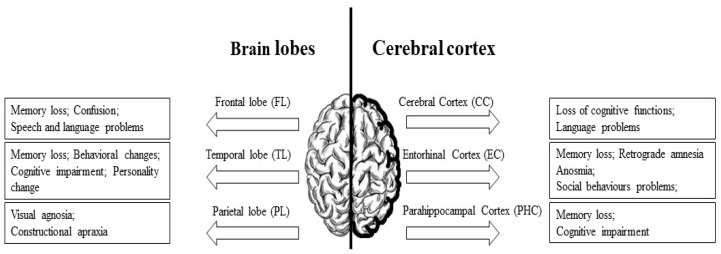
Effects of damage to individual parts of the brain throughout the course of AD.

**Figure 3 ijms-24-01059-f003:**
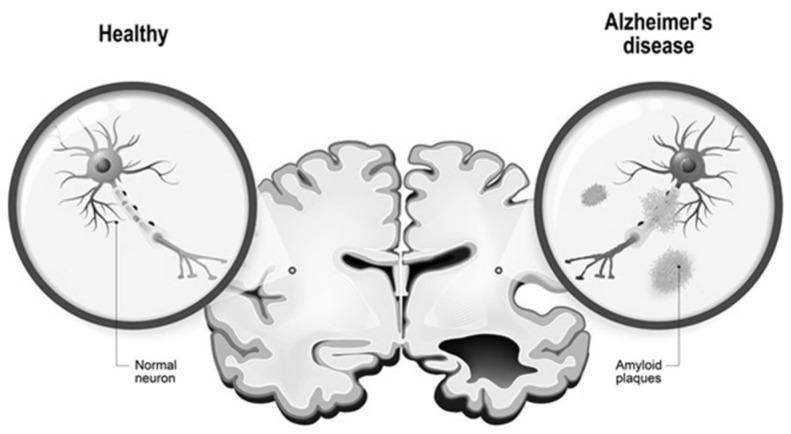
Human brain, in two halves: healthy and Alzheimer’s disease. Healthy neuron and neuron with amyloid plaques.

**Table 1 ijms-24-01059-t001:** Abnormalities in neurotransmission systems throughout the course of AD.

The System of Neurotransmission	The Area of Neurodegenerative Changes			Effect
Cholinergic	↓ Synaptic connections and neuron atrophy	Forebrain	The nucleus basalis of Meynert (NBM)	↓ Cholineacetyltransferase ↓Acetylcholine
Serotonergic		Midbrain	The raphe nuclei (brain stem)	↓ Serotonine; ↓ 5-Hydroxyindoleacetic acid
Noradrenergic		Midbrain	The locus coeruleus (brain stem)	↓ Noradrenaline
Dopaminergic		Midbrain	The brain stem	↓ Dopamine

## Data Availability

Not applicable.

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
