# Peer review of "Alzheimer’s Disease—Biochemical and Psychological Background for Diagnosis and Treatment"

_ijms, 2023, doi:10.3390/ijms24021059_

Round 1

Reviewer 1 Report

This is an interesting review covering an important area of research i.e. Alzheimer’s disease. The entire article is written carefully but it requires few modifications before it can be accepted for publication.

1.     There is only one figure in the manuscript. Figures usually attract reader’s attention and thus the authors are advised to add few relevant figures in the article.

2.     Also, the English needs improvement as there are many typo errors and erratic usage of abbreviations.

3.     The article can be benefitted from adding few relevant articles related to Alzheimer’s.

https://doi.org/10.3390/biom9090495; https://doi.org/10.3390/ijms222010986; https://doi.org/10.1016/j.molliq.2021.116888

4.     Author Contributions: 308 Funding: 309 Conflicts of Interest: All these sections are left empty.

5.      

Author Response

Thank you very much for valuable comments on the publication.

We agree with the comments of Reviewers and we corrected the manuscript „Alzheimer's disease – biochemical and psychological background for diagnosis and treatment

Reviewer # 1

  1. Two figures have been added.
  2. Languagemistakes and abbreviationshave been corrected.
  3. Suggested literature items have been added.
  4. Information about: Author Contributions, Funding: and Conflicts of Interest have been included in the new version

Reviewer 2 Report

Review on Alzheimer's disease – biochemical and psychological background for diagnosis and treatment

I have completed my review on manuscript ijms-2113267, entitled, “Alzheimer's disease – biochemical and psychological background for diagnosis and treatment.”

Alzheimer's disease (AD) is a progressive neurodegenerative disorder that causes dementia and eventually death. There is currently no efficient treatment available to slow or stop the progression of the AD. In this scenario, a review on the biochemical and psychological background for diagnosis and treatment is reasonable to support the understandings.

The topic of this review article is interesting and useful against AD. Before making a positive decision, I have some concerns and comments about the present form of the manuscript that must be addressed first.

Comments for authors

Comment 1: AD can be caused by a variety of factors. The authors' description did not provide any background information for new readers. when compared to the variety of factors that contribute to AD. All relevant information must be included in the review. This review needs to include an introduction to AD and the factors that contribute to it. Microwave exposure was also thought to be responsible for AD. I encourage authors to add some background on this topic. The suggested article may assist authors in expanding their background knowledge and understanding the mechanisms by which the EM field interacts with and affects biological systems for various effects

Article: Microwave Radiation and the Brain: Mechanisms, Current Status, and Future Prospects. International Journal of Molecular Sciences vol. 23 (2022). [https://doi.org/10.3390/ijms23169288].

Comment 2: There is no need to repeat Alzheimer's disease's full name in the text if it has already been referred to as AD. Check the manuscript again because this error appears frequently throughout the explanation.

Comment 3: The review typically included a summary of the most recent data. I suggest authors cite the most recent studies on the topic. Most of the cited information is older more than ten years.

Comment 4: In AD there is role of tau protein and ß-amyloid peptide (Aß) can you please mention some aspects about it in therapeutic prospective for easy understand to readers.

Comment 5: In line number 266 you mention about Patients with AD lose access to a coherent representation of themselves and of an object, and their life circumstances change dynamically. That indicate this disease have many dangers side effects so if you mention something about therapeutic aspects how to reduce AD effects as we know it’s not curable but at least we could reduce its effects on patient, that would be give more strong point to this review.

Comment 6. In line number 68 and 69 Laboratory diagnosis of AD should be based on direct detection of pathological changes or detections of markers in biological material showing altered expression in the early stage of the disease or which have applications in the differential diagnosis of other neurodegenerative diseases. Could you more explain about markers which are responsible in early and late-stage AD disease with some mechanism to reduce its effects.

Comment 7. The conclusion appears to be lacking and to have no key information from this review. What is the key message or advantage of this review for readers? I advise covering up the review and providing some perspective for the future based on the subject of this review.

Comment 8. What is the main advantage of this review? Why this review is important to be part of literature? What makes this review different than available literature on the subject? The abstract and conclusion must make this information clear. The significance of this review is not clear to me. I recommend to revise the abstract and conclusion.

Comment 9: There are typos and inaccuracies in the paper. I strongly recommend authors to read precisely and correct the grammatical errors and inconsistencies.

In my opinion, the manuscript should be placed on major revisions.

Author Response

Reviewer # 2

  1. This point has been exactly discussed and a suggested reference has been added:

Some factors, such as excitotoxicity, oxidative stress, cholinergic dysfunctions, tau protein hyperphosphorylation, changes in amyloid-beta peptide metabolism, apolipoprotein E, herpes viruses, insulin resistance, glycogen synthase kinase 3, and the endocannabinoid system seem to be related to pathophysiology of Alzheimer’s disease. A factor playing a role in the pathogenesis of Alzheimer's disease may also be electromagnetic radiation affecting neurotransmitters that play an important role in transmitting signals [10, 11].

2. The name Alzheimer's disease has been replaced by the abbreviation AD

3. The references have been changed according to the Reviewer's suggestions

4. This point has been exactly discussed in the new version:

In the field of research on AD a conflict has existed for many years. The amyloid hypothesis supporters argue that  and aggregation of amyloid-beta is a major factor in setting of AD. On the other hand, opponents argue that amyloid deposition is an epiphenomenon that has diverted attention away from the real causes of AD, which are still mostly unknown.  Now that amyloid-beta plaque-removing drugs for AD are available, the dispute over the amyloid hypothesis has gone from the lab to the clinic. Many amyloid-beta-targeting drugs have proved to be clinically ineffective. However FDA has granted accelerated clearance to aducanumab, one of four anti-amyloid-beta antibodies that have been demonstrated to mediate the removal of amyloid plaque from the brains of AD patients. Current therapeutics for tau treatment include targeting multiple forms of intracellular and extracellular tau to prevent pathological tau formation, accumulation, and spread. Many other drugs are in development and trials, However, experimental drug trials warrant further discussion.

5. This point has been exactly discussed in the new version:

Many activities have a beneficial effect on the mental state of the patient, reduces his depressiveness and sense of isolation, and is a source of satisfying essential needs - pleasure, joy, importance, social contacts. These include music therapy, art therapy, and exercise therapy. They are a source of stimulation and calming, increase the quality of life and are easy to use at home.

6. The third paragraph of the chapter Biochemical diagnosis of Alzheimer's disease describes this point

In recent years, the National Institute of Aging (NIA) and the International Working Group (IWG-2) have developed criteria for the early diagnosis of AD based on biomarkers and imaging data. These criteria are based on the use of a multidisciplinary approach comprising imaging modalities, clinical biochemical tests and the neuropsychological evaluation of the patient. Biochemical and imaging diagnostic criteria relate to the main features of AD – β-amyloid plaques and tau protein tangles (NFTs). A number of molecules capable of penetrating through the blood-brain barrier have been labelled which bind to Aβ or tau NFT plaques, thus allowing for the assessment of Tau aggregates in the human brain. Such a precursor molecule is Aβ-binding compound B from Pittsburgh which ushered in the era of AD neuroimaging [15, 16]. In order to facilitate interpretation of results and potential inter-facility comparison, an algorithm that enables evaluation of AD biomarkers in the cerebrospinal fluid has been created. On this scale, a score of zero (0) indicates a high probability of the absence of neurochemical disorders, while a score of 4 signals a high probability of the presence of AD

7. The above statement in the review may be a conclusion (second paragraph in the Discussion chapter):

Due to the fact that a significant increase in the incidence of AD worldwide is predicted over the next several decades, combining pharmacotherapy with a psychological approach is taken into consideration in order to extend the well-being and functionality of people diagnosed with AD.

8. This point has been exactly discussed in the new version:

This review addresses many aspects of AD, including the psychological aspect often overlooked by physicians, and the role of a person's family and caregivers in their overall well-being. This is especially important in the absence of an effective treatment for AD.

9. Language mistakes have been corrected as it was suggest

We would like to thank the Reviewers for the opinion regarding our publication.

Round 2

Reviewer 2 Report

My evaluation of the revised version is complete. All of my suggestions and queries were taken into account by the authors in the improved version. The manuscript is now suitable for publication in IJMS in my opinion.